# GIS-Based Cluster and Suitability Analysis of Crop Residues: A Case Study in Yangon Region, Myanmar

**Tin Min Htoo [1], Helmut Yabar [2,*] and Takeshi Mizunoya [2]**

[1] Degree Programs in Life and Earth Sciences, Graduate School of Science and Technology, Doctoral Program in Environmental Studies, University of Tsukuba, Tsukuba 305-8572, Japan
[2] Graduate School of Science and Technology, University of Tsukuba, 1-1-1 Tennodai, Tsukuba 305-8577, Japan
[*] Correspondence: yabar.mostacero.h.ke@u.tsukuba.ac.jp

**Abstract:** In the study of biomass assessment, geospatial modeling-based analysis becomes crucial for the sustainable management of agriculture. Currently, there is no integrated sustainability assessment of the geographic information system (GIS) cluster or suitability analysis for the feedstock of crop residues. In order to fill this research gap and support the strategy of bioenergy formulation with the circular economy concept in agriculture residues in Myanmar, this study aims to assess the energy generation potential and site locations of treatment facilities for crop residue, utilizing the integrated assessment of GIS cluster and suitability modeling. The cluster analysis identifies the rice straw as the highest feedstock of crop residues and township-based high/low clusters. In addition, the electricity generation potential is estimated at 279.14 MW for different clusters of rice straw. Moreover, the suitability analysis in the study uses the conceptual model of variables for constraints and factors with the analytical hierarchy process (AHP) technique to evaluate the weights. The suitability analysis found high suitability areas of 14,603 hectares for treatment facilities within the high/low cluster of feedstock for rice straw. The multicriteria and GIS integrated assessment model adopted in this research can support the decision-makers in developing spatial-based strategic planning for bioenergy promotion which will support sustainable farming practices in Myanmar. Additionally, the proposed model is adaptable in study areas with similar feedstock.

**Keywords:** crop residue; bioenergy; geographic information system (GIS); cluster analysis; suitability analysis; analytical hierarchy process (AHP); energy generation

## 1. Introduction

Myanmar's agriculture-based economy has resulted in an abundance of biomass resources. Every year, Myanmar produces more than 20 million tons of paddy [1]. In addition, future population growth may increase the need for food production, quickly increasing the possibility of crop residue creation. Fuelwood, wood by-products, rice husks, straw, coconut, sugarcane, palm oil, cassava, maize, manure from livestock, and poultry sector by-products are the primary biomass sources in Myanmar [2]. The traditional practices of utilizing biomass in Myanmar are cooking, heating, animal feeding, dumping, and open burning [1]. However, current management practices frequently negatively affect the environment, society, and economy. According to the studies, Asian countries accounted for one-third of global biomass greenhouse gas emissions due to the open burning of agricultural wastes [3]. Correspondingly, the Food and Agriculture Organization of the United Nations explained that Myanmar has a carbon dioxide equivalent [$CO_2$ eq] emission value of 2124.5 for crop residue [4].

According to the waste management strategy in Myanmar, one of the principles is to promote the most efficient use of resources, including resource recovery and waste avoidance. Myanmar's climate change master plan also aims to develop practices for mainstreaming GHG emission reduction into agriculture [5]. Energy utilization of residual

biomass becomes an optimal solution for sustainable waste management to develop the country's energy security as renewable energy [6]. Agricultural residues, such as rice husk and rice straw, have been used worldwide to produce renewable energy [7]. Therefore, bioenergy conversion of crop residues has become one of the effective strategies for the sustainable management of crop residues in Myanmar.

Scientific studies have demonstrated that it is possible to generate a wide variety of bioenergy and other applications from biomass residues and waste from raw goods slanted, processed, and consumed [8]. The technologies for energy conversion of biomass residue are mainly classified under thermochemical or biological methods [9]. There are some factors that influence the choice of a conversion technology to be applied to biomass in terms of quantity of the biomass feedstock, availability, choice of end-products, process economics, and environmental issues [10]. The combustion method is widely used because it is easy to implement into thermal or electrical mechanical energy.

The energy content of crop residues depends on the available feedstock of crop residue and its heating value [11]. One popular method for estimating biomass feedstock from crop residues uses the residue-to-product ratio l [12,13]. A. Chauhan et al. calculated the crop residues using crop yield and total production of crops [14]. The entire crop of straws was estimated using the economic crop yield and the straw production coefficient [15]. Jiang, Y et al. calculated the energy potential of crop residue using the official statistic of production data, residue-to-crop ratio, the lower heating value, and conversion factors [16]. Cuong et al. estimated the electricity generation of biomass using data from feedstock of crop residues, latent heat value, moisture content, and energy efficiency [17].

The spatial distribution of energy potential and feedstock crop residue can be identified using Geographic Information System (GIS). The goal of cluster analysis is to determine if a dataset comprises a single group or contains subgroups to select how many clusters are in the data [18,19]. M. Svazas et al. considered that reducing communication and transportation expenses is the primary justification for clustering because geographic proximity enables businesses to access a big pool of suppliers [12]. For GIS cluster mapping, V. Prasannakumar et al. used the ArcGIS tools of Spatial Autocorrelation Moran's I method, and Getis-Ord GI* function and density estimation using Kernel density tool to find the road accident hotspots [20]. K. M. Dao et al. applied ArcGIS in terms of the "Calculate Distance Band from Neighbor Count", "Incremental Spatial Autocorrelations", and the "Getis-Ord GI*" function to make the clusters of pig farms [21]. F. Venier and H. Yabar grouped cattle farms according to their sizes in terms of large, medium, and minor scales using GIS [22].

Globally, to achieve the goal of sustainable renewable energy development, most countries worldwide have worked hard at a regional and global level by constructing biomass treatment facilities using spatial analysis [23]. Therefore, geospatial consideration and location modeling are required for a suitable location of treatment facilities' site selection and analysis. Another strand looks at land suitability analysis and site selection for urban and rural agriculture using a variety of methodologies, such as multicriteria decision-making (MCDM) [24].

MCDM approaches are utilized in site selection applications to determine the weights of defined main/sub-criteria [25]. R. Rios and S. Duarte considered the thirty-three constraints and seven criteria factors for GIS suitability analysis [26]. E. C. Chukwuma et al. studied the suitability map for biogas power plants using the constraints of rural and urban areas, park and recreational areas, water bodies, wetlands, roads, transmission lines, and slopes [23]. L. Jayarathna et al. considered the criteria factor in terms of vegetation cover, crop cover, slope, road proximity, transmission line proximity, workforce potential, energy demand potential, population exposure, and viewshed to locate the biomass energy plant [27]. Nantasaksiri, K et al. use the analytical hierarchy process (AHP) of multicriteria decision-making to evaluate the weight of the factors [28].

Currently, there is no integrated assessment of the GIS cluster and site suitability analysis for biomass energy potential. Moreover, to support the strategy of bioenergy

formulation with the circular economy concept for agriculture residues in Myanmar, this study aims to assess the energy generation potential of crop residue in terms of the township based spatial high/low clusters and to find suitable sites for treatment facilities for energy generation.

## 2. Materials and Method

### 2.1. Study Area

The study area comprises 43 townships in the Yangon Region, which is in the delta area regarding the ecological environment. The selected area is examined to assess the potential locations of treatment facilities based on available clusters of agro residues, refer to Figure 1. In 2019, the Yangon region had a population of over eight million [29] and covered an area of 10,171 square kilometers [30]. It is one of the highest-production areas of rice in Myanmar [31]. Additionally, the major crops in the region are paddy, green gram, groundnut, sunflower, sesame, and sugarcane [32]. Regarding meat from livestock production, cattle, buffalo, pigs, chickens, ducks, and quails are primary sources of meat production [33].

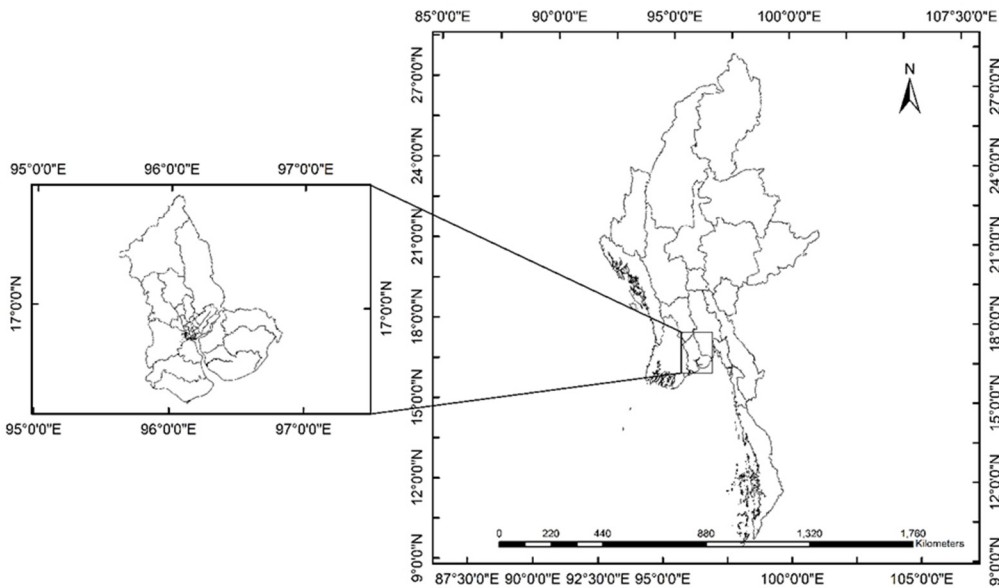

**Figure 1.** Study Area of Yangon Region, Myanmar.

### 2.2. Research Flow and Data Collection

This study used ArcGIS Desktop 10.8.1 (Product Version: 10.8.1.14362), developed by Environmental Systems Research Institute (Esri), a company based in California, United States, to perform a GIS cluster and suitability analysis, as shown in Figure 2.

This study has two categories of data: nonspatial and spatial. The nonspatial data regarding the production number of crop output quantity, electricity access, and water access are collected from Myanmar Information Management Unit [33,34]. The spatial data in land use and land cover maps, digital elevation model (DEM), road, protected areas, and administrative data are gathered from open satellite images of open data sources from the United States Geological Survey(USGS), Environmental Systems Research Institute (Esri) 2020 landcover and Geofabrik website and Myanmar Information Management Unit website, Table A1, Appendix A.

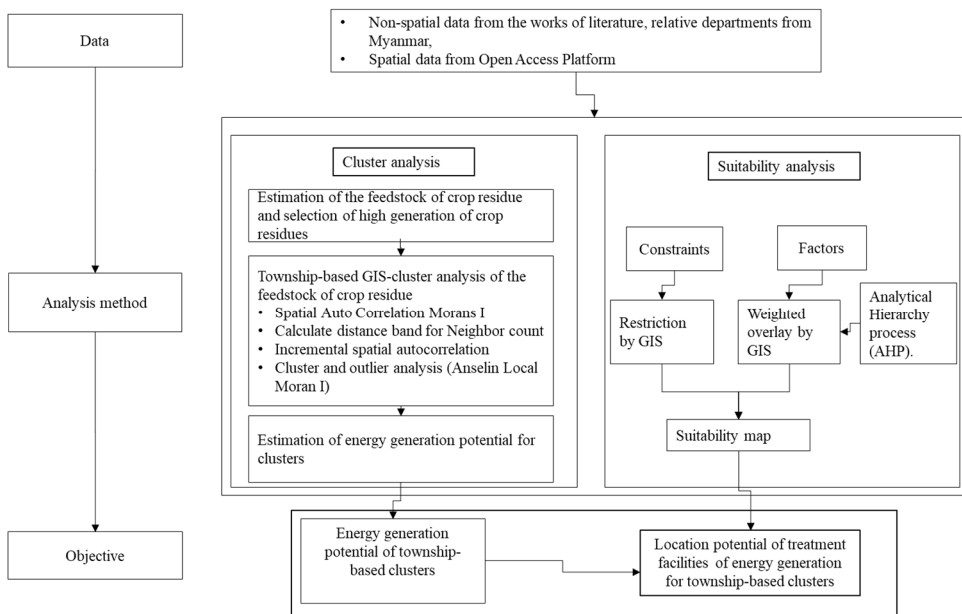

**Figure 2.** Research Framework.

*2.3. Cluster Analysis*

Cluster analysis is divided into three components, as follows:

a.   Calculating feedstock of crop residue and selecting of targeting biomass waste type
b.   Township-based cluster analysis of feedstock using ArcGIS
c.   Estimating the electricity generation potential of each cluster

2.3.1. Calculation of Crop Residue

Equation (1) computes the feedstock for the crop residue potential. The RPR values of different crops are shown in Table A2; Appendix A. Grain production data are collected from township profiles from the Myanmar Information Management Unit. The data are processed to connect the attribute table of the vector file of the townships in the study region in the ArcGIS 10.8.1 environment after calculating the biomass waste generation potential from crop residue, as indicated in Equations (1) below:

$$\sum_{i=1}^{n} Gi * RPRi \tag{1}$$

where,

*Gi* = Grain production
*RPRi* = residue to crop ratio

Thus, waste generation potential is equal to total grain production given by (*Gi*) times residue to crop ratio denoted by (*RPRi*).

2.3.2. Township-Based GIS Cluster Analysis of the Feedstock of the Crop Residue

After calculating the crop residues in the 43 townships, the selected highest potential of crop residues generation data is prepared in ArcGIS to make the township-based cluster analysis with four steps. In step one, the Spatial Autocorrelation (Global Moran's I) tool simultaneously measures spatial autocorrelation based on selected crop residue generation amounts and township-based locations. It evaluates whether the pattern expressed is clustered, dispersed, or random using Moran's I Index value of both the z-score and *p*-value. The z-score and *p*-value indicate statistical significance. Step two is to determine the average distance of neighbor townships using the tool Calculate Distance Band from Neighbor Count. In step three, the tool of Incremental Spatial Autocorrelations creates a line graph and finds the distances where township-based spatial clustering patterns

occurred with corresponding statistically significant peak z-scores. These peak distances are used as a radius parameter to ensure step four of Cluster and Outlier Analysis (Anselin Local Moran I). It distinguishes between a statistically significant township-based cluster of high values (High-High Cluster), clusters of low values (Low-Low Cluster), outliers in which a high value is surrounded primarily by low values (High-Low Cluster), and outliers in which a low value is surrounded primarily by high values (Low-High Cluster).

### 2.3.3. Estimating the Electricity Generation Potential of Each Cluster

The electricity generation potential for the township based on high/low clusters was estimated using Table 1.

**Table 1.** Electricity Potential Calculation Method [17].

| |
|---|
| $Ep(MW) = \frac{Rice\ straw\ generation*(1-LRS)*(1-MC)*LHV*\mu}{3.6*OPH}$ |
| *EP* = Electricity potential |
| *LRS* = Loss of rice straw during handling and storage (%) = 10 |
| *MC* = Moisture content assumed on a dry basis (%) = 12 |
| *LHV* = Low heating value of residue (MJ per kg) = 14 |
| *μ* = Overall efficiency of the plant (%) = 25 |
| *OPH* = Operation hours = 8000 |

### 2.4. Suitability Analysis

In this study, the suitability analysis model Equation (2) is for the locations of treatment facilities for electricity generation of the township-based high/low clusters. The treatment facilities were modeled to reduce the environmental, economic, and social negative impacts on the whole Yangon region. Suitability analysis for GIS-MCDM encompassed two paradigms with the conceptual model where constraints variables and factors variables [35]. In addition, the evaluation of weights for factor variables was analyzed using the multicriteria decision-making method in the analytical hierarchy process (AHP).

$$S = \sum_{i=1}^{n} f_i w_i * \prod_{j=1}^{n} c_j \qquad (2)$$

*S* = suitability
$f_i$ = factors variables
$w_i$ = weight of factors variables
$c_j$ = constraints variables
*i* = different restrictions for constraints represented by the individual parameters

### 2.4.1. Constraints Variables

A restriction serves to limit the options under consideration [35]. The literature study investigated the buffer criteria and distance in Table A3, Appendix B, for restrictions. After processing buffer and union in ArcGIS, the criteria are changed to raster. Finally, using ArcGIS raster calculator, all constraint variables are combined as a restriction map of the Yangon Region.

### 2.4.2. Factor Variables

A weighted linear combination combines factors variables by assigning each a weight and then summing the results to obtain a suitability map [35]. Factors of the criteria and preference distance are shown in Table A4, Appendix B. The evaluation of weights is calculated from the Multicriteria Decision Making Method. Caprioli, C. and Bottero, pointed out that different multicriteria techniques are used to solve site selection problems [36]. These include the Preference Ranking Organization Method for Enrichment of Evaluations (PROMETHEE), ÉLimination Et Choix Traduisant la REalité (ELECTRE), techniques for

order preference by similarity to ideal solution (TOPSIS), and Analytic Hierarchy Process (AHP). This paper uses a weight evaluation method based on the AHP methodology. First, criteria factors are processed using Euclidean Distance and Reclassification techniques in Spatial Analyst Tools of ArcGIS 10.8.1, and the weighted overlay is processed using the weighted value from the AHP of MCDM analysis to generate the final suitability map.

The analytical hierarchy process (AHP) is a general theory of measurement to derive ratio scales from discrete and continuous paired comparisons [37]. This study applied it in Table A5, Appendix B. First, the consistency of the matrix data is defined and measured by a formula employing the average of the nonprincipled eigenvalues, which is then compared to other values in a numerical scale ranging from 1 to 9. Since the Consistency Ratio (C.R.) is acceptable at less than 0.1, the evaluated weight of each Factor of AHP is overlayed using the tools of weighted overlay in ArcGIS. Then the restriction map and factor map are combined using the Times of Spatial Analyst Tool of ArcGIS tools to find the final suitability map.

## 3. Results

### 3.1. Cluster Analysis

#### 3.1.1. Crop Residue Generation

The information on crop residue generation from 43 townships is shown in Figure 3, and further details are highlighted in Table A6, Appendix C. According to the results of the crop residue generation in Figure 3, rice straw generation amounts to 2,900,157.63 tons/year, which has the highest potential compared to the other 18 types of resides. Therefore, rice straw is selected as the targeted crop residue for township-based clusters in ArcGIS 10.8.1.

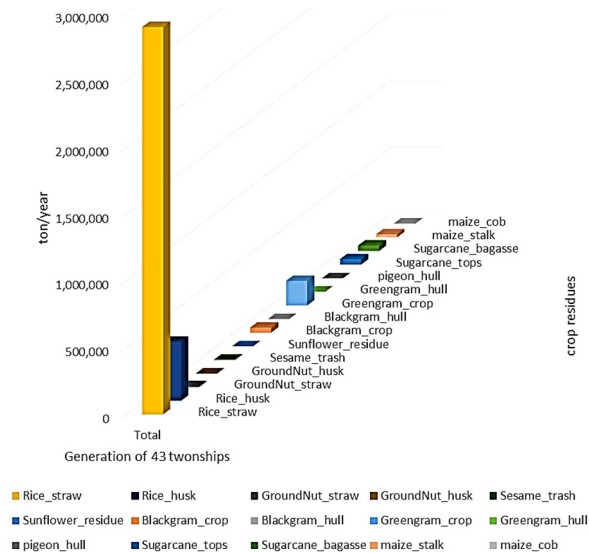

**Figure 3.** Crop Residue Generation from 43 Townships.

#### 3.1.2. Township-Based Clusters Analysis of the Feedstock of the Rice Straw

According to step one, "Spatial Autocorrelation (Global Moran's I)", the z-score is 6.56, and the *p*-value is less than 10% Figure 4. The result indicates that the cluster pattern exists in 43 townships. Then, step two of the tool "Calculate Distance Band from Neighbor Count" finds 6948.84 m as the average distance among 43 townships. Then, step three of "Incremental Spatial Autocorrelations" ensure the result in the peak distance of 20,846.51 m as the radius parameter that promotes the township-based cluster Figure 5. Finally, step four distinguishes township-based clusters of high values (High-High Cluster), clusters of low values (Low-Low Cluster), outliers in which a high value is surrounded primarily by low values (High-Low Cluster), and outliers in which a low value is surrounded primarily by high values (Low-High Cluster) Figure 6, Table 2.

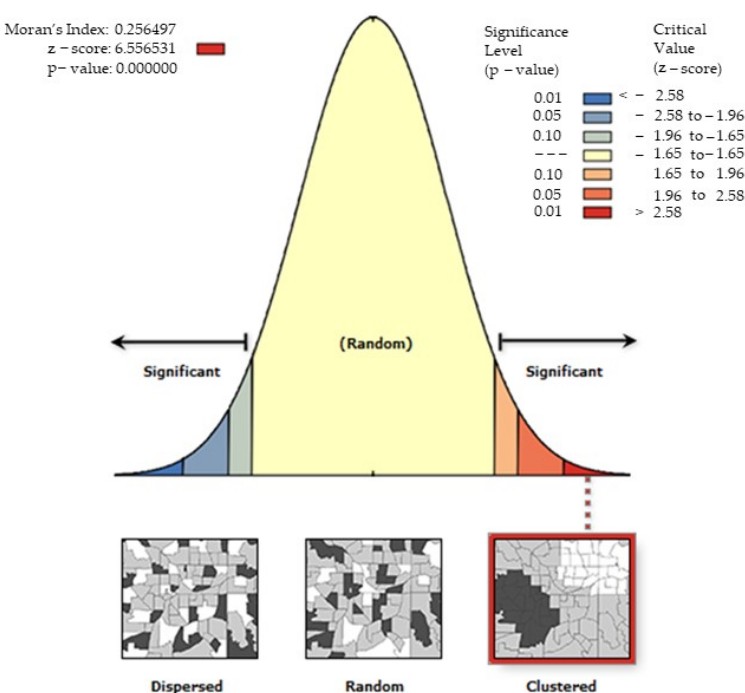

**Figure 4.** The result of Spatial Autocorrelation (Global Moran's I).

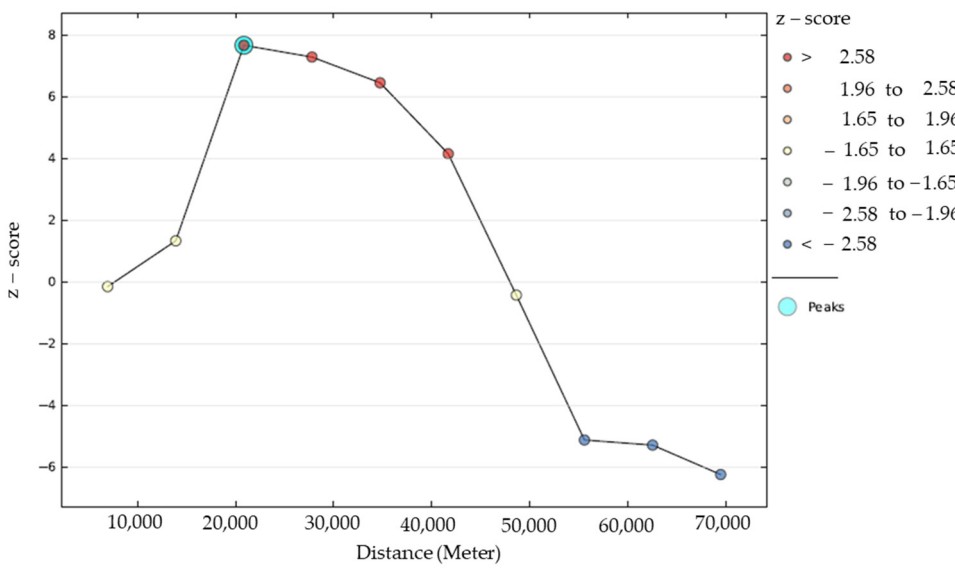

**Figure 5.** The result of Incremental Spatial Autocorrelations.

**Table 2.** The Results of Township-Based Clusters Analysis of the Feedstock of the Rice Straw.

| Township Name | Cluster Type | Rice Straw Generation (ton) | Electricity Potential (MW) |
|---|---|---|---|
| Kungyangon | Not Significant | 285,000.43 | 27.43 |
| Thongwa | Not Significant | 381,007.22 | 36.67 |
| Hmawbi | Not Significant | 11,308.38 | 1.09 |
| Hlegu | Not Significant | 292,719.14 | 28.17 |
| Twantay | Not Significant | 291,133.19 | 28.02 |
| Shwepyithar | Not Significant | 274.16 | 0.03 |
| Taikkyi | Not Significant | 170,932.38 | 16.45 |

**Table 2.** *Cont.*

| Township Name | Cluster Type | Rice Straw Generation (ton) | Electricity Potential (MW) |
|---|---|---|---|
| Kyauktan | Not Significant | 262,814.46 | 25.30 |
| Htantabin | Not Significant | 355,211.87 | 34.19 |
| Kawhmu | High-High Cluster | 261,696.52 | 25.19 |
| Kayan | High-High Cluster | 277,279.32 | 26.69 |
| Dala | High-Low Cluster | 110,021.13 | 10.59 |
| Thanlyin | High-Low Cluster | 157,252.94 | 15.14 |
| Dagon Myothit (East) | Low-Low Cluster | 22,458.52 | 2.16 |
| Dagon Myothit (Seikkan) | Low-Low Cluster | 15,501.94 | 1.49 |
| Mingaladon | Low-Low Cluster | 1207.23 | 0.12 |
| Kyeemyindaing | Low-Low Cluster | 3637.47 | 0.35 |
| Seikgyikanaungto | Low-Low Cluster | 701.33 | 0.07 |

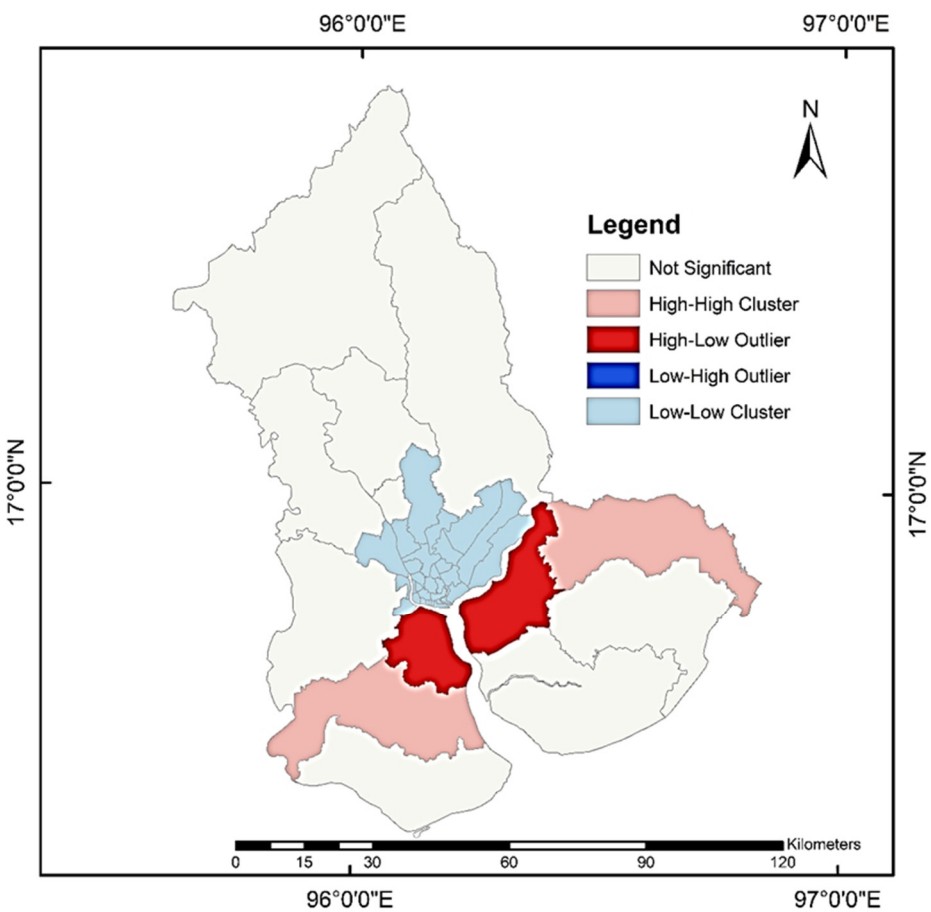

**Figure 6.** Township-based clusters for rice straw.

3.1.3. Estimating the Electricity Generation Potential of Each Cluster

According to the electrical potential of the different township-based clusters in Table 2, the total energy potential into the national grid from rice straw within the 43 townships is 279.14 MW. High-High Clusters have 51.88 MW, while High-Low Clusters have 25.73 MW. Low-Low Clusters have 3.84 MW. Although some of the townships do not exit as cluster patterns, Kungyangon, Thongwa, and Hmawbi have the potential of 27.43 MW, 36.67 MW,

and 28.17 MW, respectively. Applying the electric potential from different clusters, the decision maker can decide on the strategic plan for bioenergy promotion.

### 3.2. Suitability Map for Yangon Region

All constraint variables are mapped in Figure A1, Appendix B. The restriction areas of all constraint variables are identified in Figure 7. The restricted areas are represented by green and unrestricted, shown in blue in the Yangon Region. The AHP method gives the results of the evaluated weight of each factor variable in Table 3. For all factor variables, maps are reclassified, Figure A2, Appendix B. The result of the overlays of all factor variables as the preferred areas includes low, intermediate, and high suitability, as in Figure 8. After combinations of variables for constraints and factors, the final suitability map results in terms of no suitability, low suitability, intermediate suitability, and high suitability Figure 9. The resulting areas of the suitability map are classified as follows, low suitability (26,222 hectares), intermediate suitability (111,799 hectares), high suitability (14,603 hectares), and the total available land equivalent to 152,624 hectares. The high suitability areas are identified as the final suitable areas for treatment facilities of electricity generation of township-based clusters.

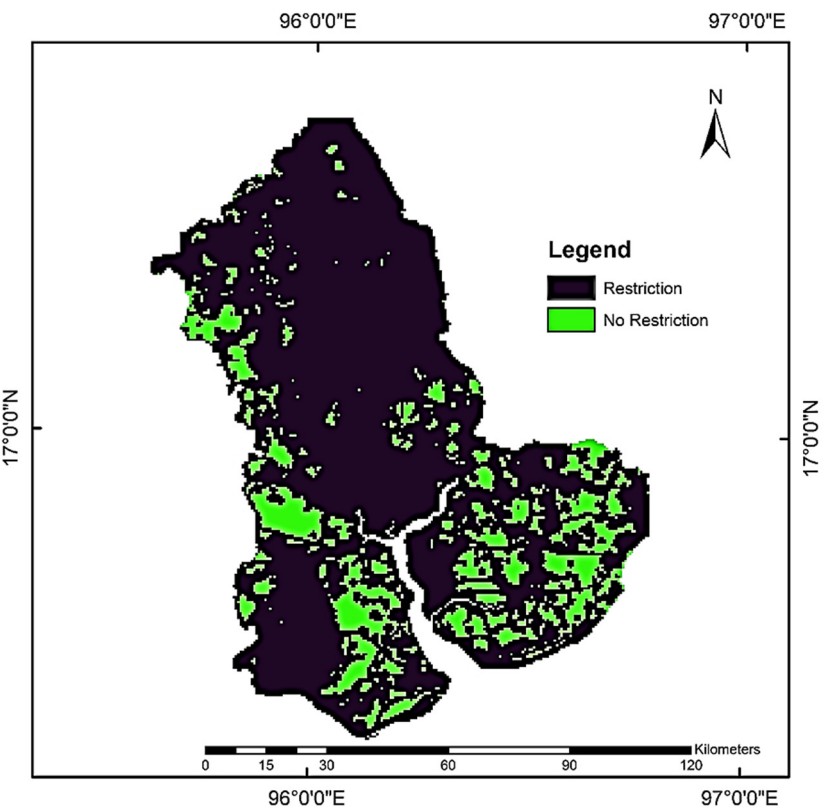

**Figure 7.** Constraint Variables for Restriction Map.

**Table 3.** Evaluated Weight by AHP Method.

| Method | Crop and Livestock | Residential Industry | | Road | Railway | Waterway | Slope | Electricity Access | Water Supply | Earthquake Faults | Flood Zone |
|---|---|---|---|---|---|---|---|---|---|---|---|
| Weight | 0.14 | 0.14 | 0.14 | 0.14 | 0.02 | 0.02 | 0.14 | 0.08 | 0.04 | 0.05 | 0.10 |
| λmax | 11.45 | | | | | | | | | | |
| Consistency Index (CI) | 0.04 | | | | | | | | | | |
| RI | 1.51 | | | | | | | | | | |
| CR | 0.02 | | | | | | | | | | |

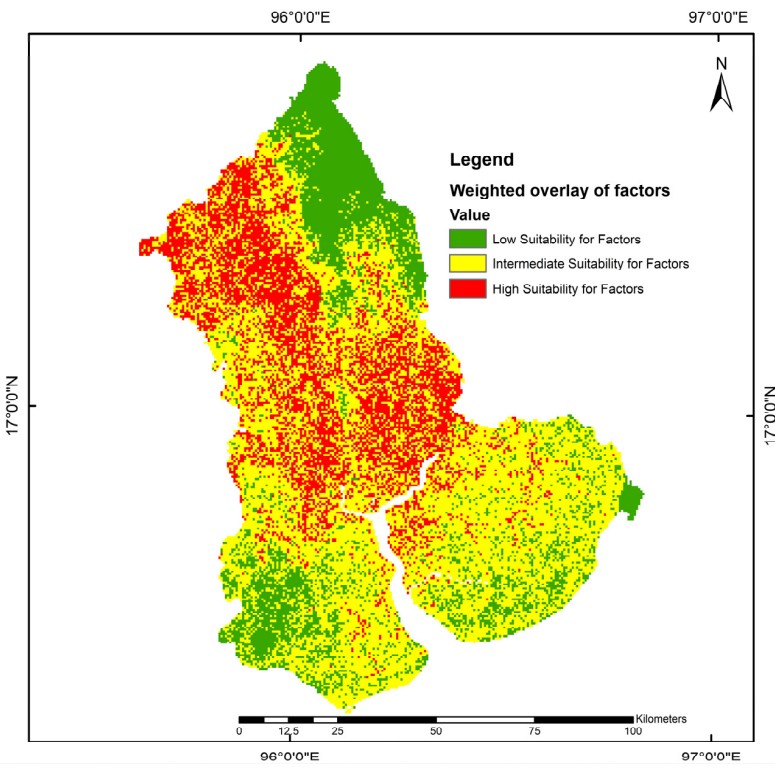

**Figure 8.** Factor variables for Weighted Overlay Map.

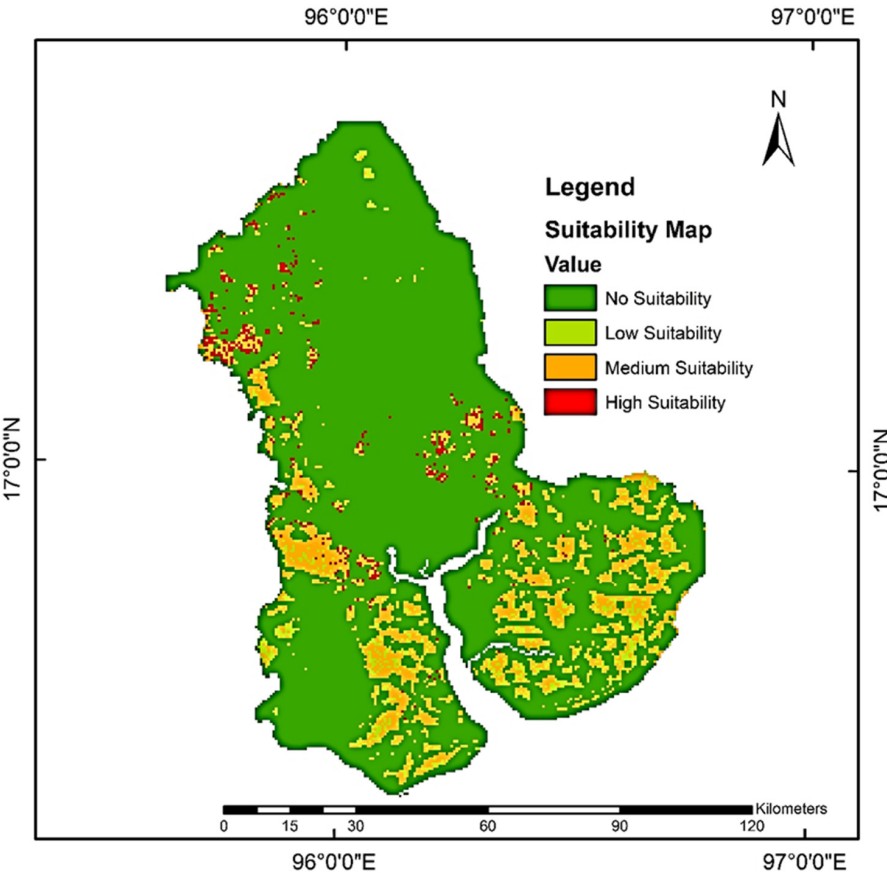

**Figure 9.** Final Suitability Map for Treatment Facilities.

### 3.3. Site Suitability Locations of Treatment Facilities for Township-Based Clusters

In ArcGIS, township-based clusters are processed to make the intersection with the final suitable areas Figure 10. For the High-High Cluster, the intersection finds 11 areas of polygons with a total area of 292 ha. In addition, there are 19 polygons with 642 ha for the High-Low Cluster, 10 with 334 ha for the Low-Low Cluster, and 229 with 12,358 ha for not significant clusters. This result gives the information to decision-makers to select the strategic locations of treatment within the different clusters.

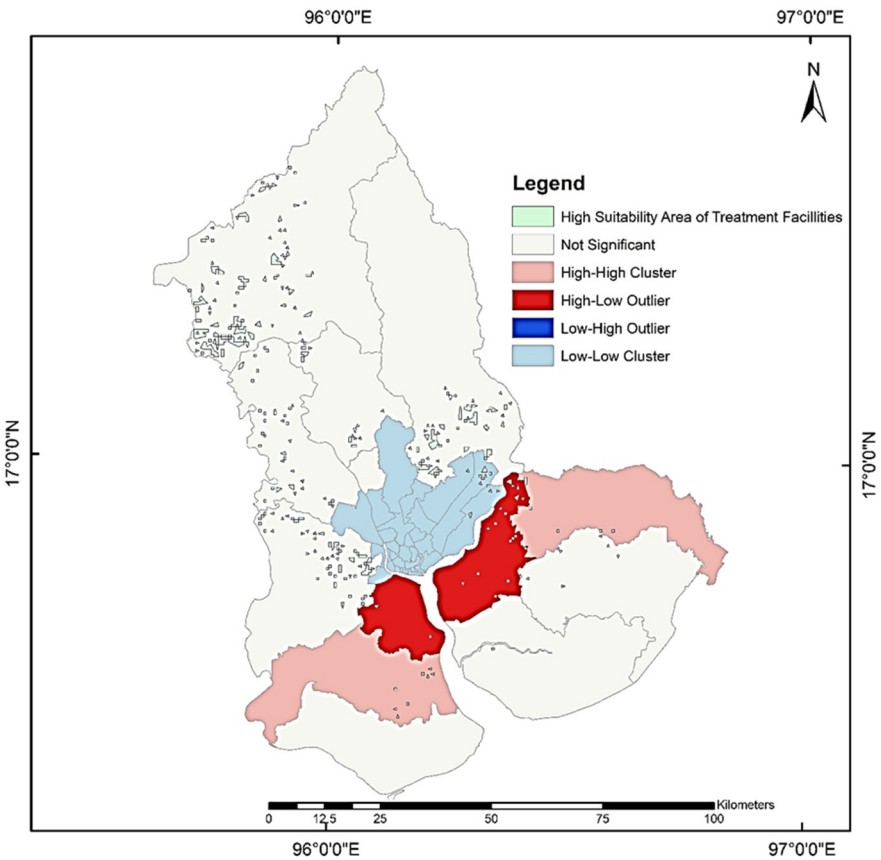

**Figure 10.** Township-based Clusters and Locations of Treatment Facilities.

## 4. Discussion

Under Cluster Analysis, the evaluation of crop residue is based on the yearly production rate (secondary data collected from 43 townships in the Yangon Region, Myanmar). Residue to production ratio RPR and crop production are used to calculate the generation of crop residues. In this study, the production value is collected from the data for 2019 of the general administrative office, which is on the Myanmar Information Management Unit website. Moreover, RPR values are gathered from different countries and different pieces of literature. The disadvantage of this method using RPR values is that different crops may have varied RPR values depending on weather variations, crop types, water availability, soil quality, and farming methods [38]. Therefore, estimating the number of residues using an RPR may result in inaccurate estimates. Thus, the more accurate generation calculation should also depend on the more reliable statistical production data and precise testing results RPR might improve.

Another way to calculate the crop residue which uses crop production is based on the cropped area and is frequently used for woody residues from perennial crops [38]. This approach assumes that the amount of crop and residue harvested from a cropping area can vary significantly depending on the management style (traditional or advanced) and crop variety. The remote sensing method using satellite images is also an efficient

and cost-effective way to obtain information concerning crop residue and crop condition cover [39].

In GIS cluster analysis, there are four steps in terms of Spatial Auto Correlation Morans I; Calculate Distance Band for Neighbor Count; Incremental Spatial Autocorrelations, and Cluster and Outlier Analysis (Anselin Local Moran I) in this study. However, instead of action four of Cluster and Outlier Analysis (Anselin Local Moran I), Hot Spot Analysis (Getis-Ord Gi*) can also be used to identify the extraordinary and hot spots.

In GIS suitability analysis, the constraint variables and the factor variables are modeled. When determining whether the land has the potential to meet the criteria, there is no set standard for the assessment of conditions and factors or other considerations. The constraints depend not only on the regulations but also on the sustainability consideration. In this study, evaluating the weight of factors is the AHP method. For factors, the different multicriteria decision-making (MCDM) can be used to solve site selection problems of weighting in terms of preference ranking organization method for enrichment evaluation (PROMETHEE), ÉLimination Et Choix Traduisant la REalité (ELECTRE), Technique for Order of Preference by Similarity to Ideal Solution (TOPSIS), and others.

GIS cluster and suitability analysis need vector spatial data that can be classified from a satellite image or related department. In this study, land use and land cover data are mainly from the Environmental Systems Research Institute (Esri) 2020 land cover, which has an accuracy of 87 percent. The other data are downloaded from the Myanmar Information Management Unit.

In this study, The GIS cluster analysis can identify the rice straw as the promising feedstock among crop residues. Furthermore, the township-based clusters are helpful for strategic planning and the suitable location of treatment facilities. Therefore, connecting the cluster of biomass waste with viable facilities should be studied in the future.

## 5. Conclusions

In this study, the rice straw with the most significant potential is 2,900,157.63 tons/year. The total electricity potential from the rice straw of the Yangon region to the national grid is 279.14 MW. The high/high clusters of Kawhmu and Kayan are suitable townships to promote the bioenergy development plan because these townships are surrounded by high feedstock potential for rice straw. The electric potential of rice straw from Kawhmu and Kayan township is 51.88 MW. Moreover, the suitable locations of treatment facilities for different clusters have 269 polygons with a total area of 13,626 ha. For generating scenarios for optimal biomass management system planning in terms of the transportation network, which is a crucial aspect of feedstock management, it is required to apply the results in this study. For recommendation, the future study should focus on allocating the resource clusters and treatment facilities in road networks and household applications. Moreover, life cycle assessment can be used to assess the performance of GHG emission reduction potentials from different scenarios.

**Author Contributions:** Conceptualization, H.Y.; Methodology, T.M.H., H.Y. and T.M.; Formal analysis, T.M.H.; Investigation, H.Y.; Writing – original draft, T.M.H.; Writing – review & editing, H.Y. and T.M.; Supervision, H.Y. and T.M. All authors have read and agreed to the published version of the manuscript.

**Funding:** This research received no external funding.

**Institutional Review Board Statement:** Not applicable.

**Informed Consent Statement:** This does not apply to the study.

**Data Availability Statement:** All data are reported in this work.

**Conflicts of Interest:** The authors declare no conflict of interest.

## Appendix A

**Table A1.** Spatial and Non-Spatial Data.

| Data | References |
|---|---|
| Residentials, crops, forests water body | Esri land cover map [40] |
| Digital Elevation Model (DEM) | USGS/ Earth Explorer/ SRTM/SRTM 1 arc-second Global [41] |
| Waterbody | USGS/ Earth Explorer/ SRTM/SRTM waterbody data [41] |
| Yangon townships boundary map | [42] |
| Road network | [43,44] |
| River network | [44] |
| Railway | [45] |
| Airport | [46] |
| Industrial zone from township data | Georeferencing in Arc Map 10.8.1 using maps from Township Profiles [35] |
| Protected and heritage areas | [47] |
| Flood zone potential | [48] |
| Earthquake faults | [49] |
| Yangon Region landuse, Grain production data from township profiles in Yangon Region, Utilities access data | [43,50] |
| Medium voltage transmission line | [51] |
| High voltage transmission line | [52] |
| Power station | Google Earth |

**Table A2.** Type of Crop Residue and RPR Value.

| Crop Residue | Residue-to-Crop Ratio | Reference |
|---|---|---|
| Rice straw | 1.757 | [53] |
| Rice husk | 0.267 | [53] |
| Ground nut straw | 2.3 | [54] |
| Ground nut husk | 0.477 | [38] |
| Sesame trash | 2 | [55] |
| Sunflower trash | [0.7–3.5] used as 1.5 | [56] |
| Black gram straw | 1.7 | [57] |
| Black gram husk | 0.03 | [57] |
| Green gram straw | 1.7 | [57] |
| Green gram husk | 0.03 | [57] |
| Pigeon bean straw | 1.7 | [57] |
| Pigeon bean husk | 0.03 | [57] |
| Sugarcane top | 0.3 | [58] |
| Sugarcane bagasse | 0.299 | [56] |
| Maize straw | (0.7–2.5) used as 2 | [56] |
| Maize husk | 0.273 | [53] |

## Appendix B

**Table A3.** The Criteria of Constraint Variables and Buffer Distance.

| Constraint Variables | References | | | | | | | This Study |
|---|---|---|---|---|---|---|---|---|
| | [23] | [59] | [60] | [61] | [62] | [63] | [64] | |
| Airports | | Within 8 km | | | | 7–12 km | | 7 km [63] |
| Environmentally sensitive areas [forest and mangrove] | 500 m | 1 km | | | | 2 km | | 1 km [59] |
| Commercial and recreational areas | 500 m | 500 m | | | | | | 500 m [59] |
| Roads | | 300 m | | | | | | 300 m [59] |
| Power stations | | 100 m | | | | | | 100 m [59] |
| Transmission lines [H.V.] | 100 m | 100 m | | | | | | 100 m [59] |
| Slope | 15 | slopes >15% | | | | | | 15% [59] |
| Residentials | 1 km | 1 km | | >20 km | | | 0–50 m | 1 km [59] |
| Waterbody | 200 m | 300 m | Lowest >220 m Highest 0–2 km | Lowest 200 m Highest >10 km | 500 m | 1000 m | 500 m | 500 m [62] |
| Culture important places | | | | | | 5 km | | 5 km [63] |
| Road | 30 m | | | | 1 km | | 500 m | 500 m [64] |

**Table A4.** The Criteria of Factors Variables and Preference Distance.

| Factor Variables | Reference | | | | | | This Study |
|---|---|---|---|---|---|---|---|
| | [59] | [60] | [61] | [62] | [63] | [64] | |
| Crop land | | | | | | | >500 m |
| Residentials | Lowest <1000 highest >4000 | Lowest 0–100 Highest > 500 | Lowest 0–1.5 km Highest >12 km | | 500–2000 m | Highest >50 | >500 m [63] |
| Manufacture | | | | | | | >500 m |
| Road | Lowest >2000 Highest <200 | Lowest 2 km Highest 200 | Lowest −300 m Highest >10 km | <1 km | 1500 m | 1601–2000 m Highest 501–1000 | >200 m [60] |
| Railway | | Lowest > 20,000 Highest 0–2000 | | | 100–500 m | | >500 m [63] |
| Waterway | | | | | | | >500 m |
| Slope | | | Lowest >20 Highest 0–6 | | | Lowest >30 Highest 0–10 | >10 [64] |
| Electricity access | | | | | | | Priority on the area where firewood is used [65] |
| Water supply access | | | | | | | Priority on tube well use [65] |
| Earthquake faults | | | Lowest 0–2 km Highest >20 km | | | | >2 km [61] |
| Flood zone | | | | | | | >2 km |

**Table A5.** Pairwise Comparison Matrix of Factors.

| | Crop and Livestock | Residential | Industry | Road | Railway | Waterway | Slope | Electricity Access | Water Supply | Earthquake Faults | Flood Zone |
|---|---|---|---|---|---|---|---|---|---|---|---|
| Crop | 1.00 | 1.00 | 1.00 | 1.00 | 7.00 | 7.00 | 1.00 | 2.00 | 3.00 | 3.00 | 2.00 |
| Residential | 1.00 | 1.00 | 1.00 | 1.00 | 7.00 | 7.00 | 1.00 | 2.00 | 3.00 | 3.00 | 2.00 |
| Industry | 1.00 | 1.00 | 1.00 | 1.00 | 7.00 | 7.00 | 1.00 | 2.00 | 3.00 | 3.00 | 2.00 |
| Road | 1.00 | 1.00 | 1.00 | 1.00 | 7.00 | 7.00 | 1.00 | 2.00 | 3.00 | 3.00 | 2.00 |
| Railway | 0.14 | 0.14 | 0.14 | 0.14 | 1.00 | 1.00 | 0.14 | 0.13 | 0.20 | 0.14 | 0.14 |
| Waterway | 0.14 | 0.14 | 0.14 | 0.14 | 1.00 | 1.00 | 0.14 | 0.20 | 0.33 | 0.33 | 0.20 |
| Slope | 1.00 | 1.00 | 1.00 | 1.00 | 7.00 | 7.00 | 1.00 | 3.00 | 5.00 | 3.00 | 2.00 |
| Electricity Access | 0.50 | 0.50 | 0.50 | 0.50 | 8.00 | 5.00 | 0.33 | 1.00 | 3.00 | 2.00 | 0.33 |
| Water supply | 0.33 | 0.33 | 0.33 | 0.33 | 5.00 | 3.00 | 0.20 | 0.33 | 1.00 | 0.50 | 0.20 |
| Earthquake Faults | 0.33 | 0.33 | 0.33 | 0.33 | 7.00 | 3.00 | 0.33 | 0.50 | 2.00 | 1.00 | 0.33 |
| Flood zone | 0.50 | 0.50 | 0.50 | 0.50 | 7.00 | 5.00 | 0.50 | 3.00 | 5.00 | 3.00 | 1.00 |
| SUM | 6.95 | 6.95 | 6.95 | 6.95 | 64.00 | 53.00 | 6.65 | 16.16 | 28.53 | 21.98 | 12.21 |

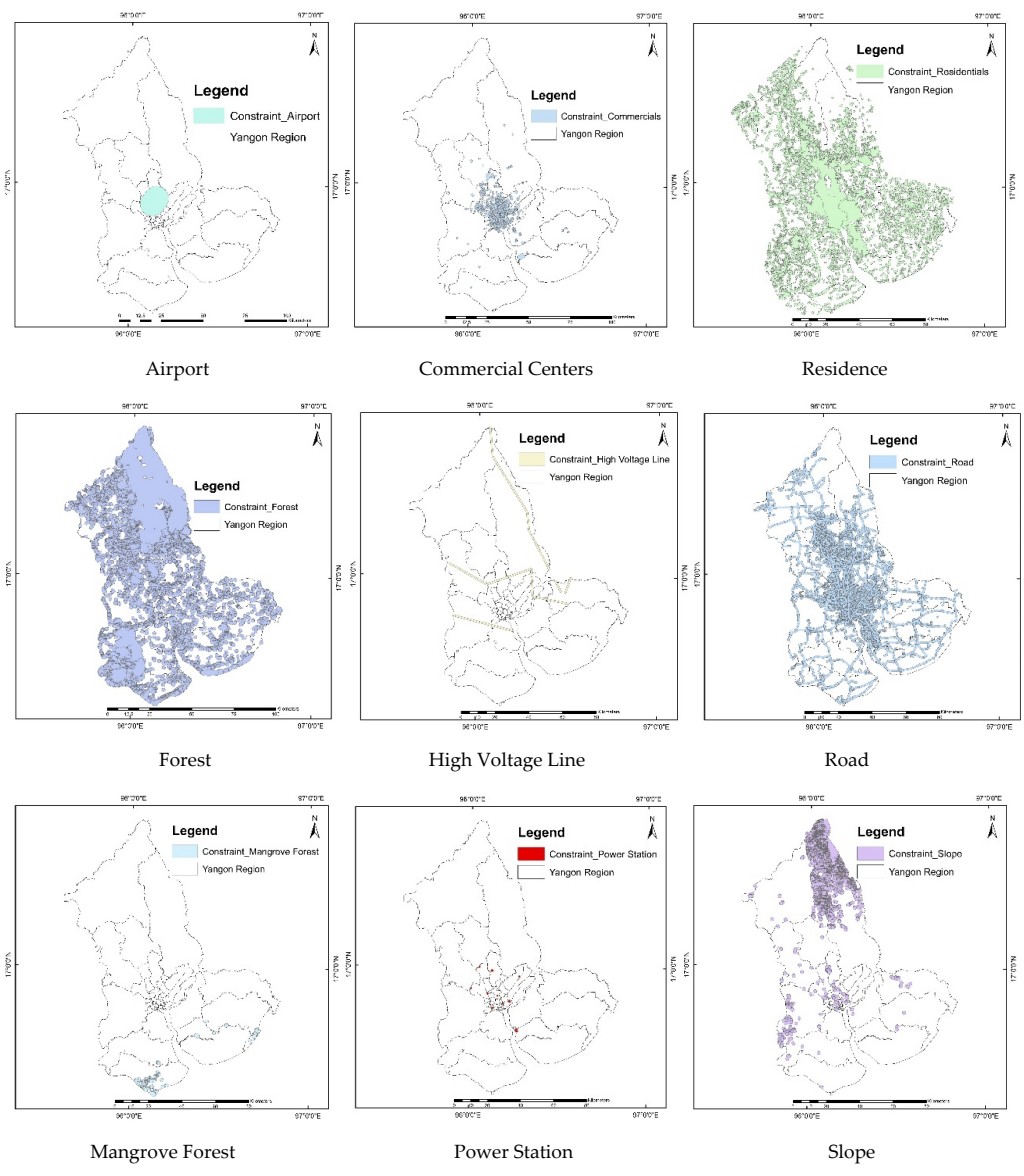

Airport     Commercial Centers     Residence

Forest     High Voltage Line     Road

Mangrove Forest     Power Station     Slope

**Figure A1.** *Cont.*

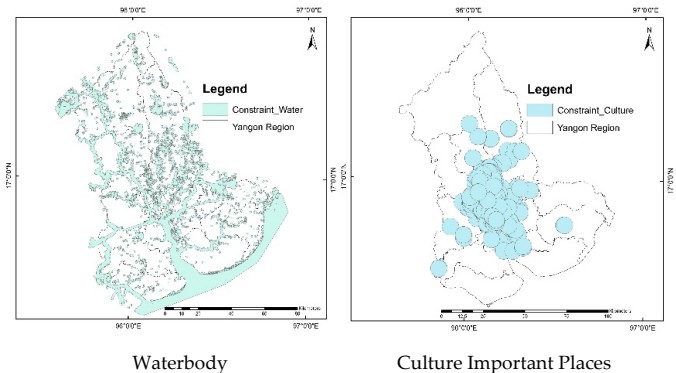

Waterbody                    Culture Important Places

**Figure A1.** Constraints Map.

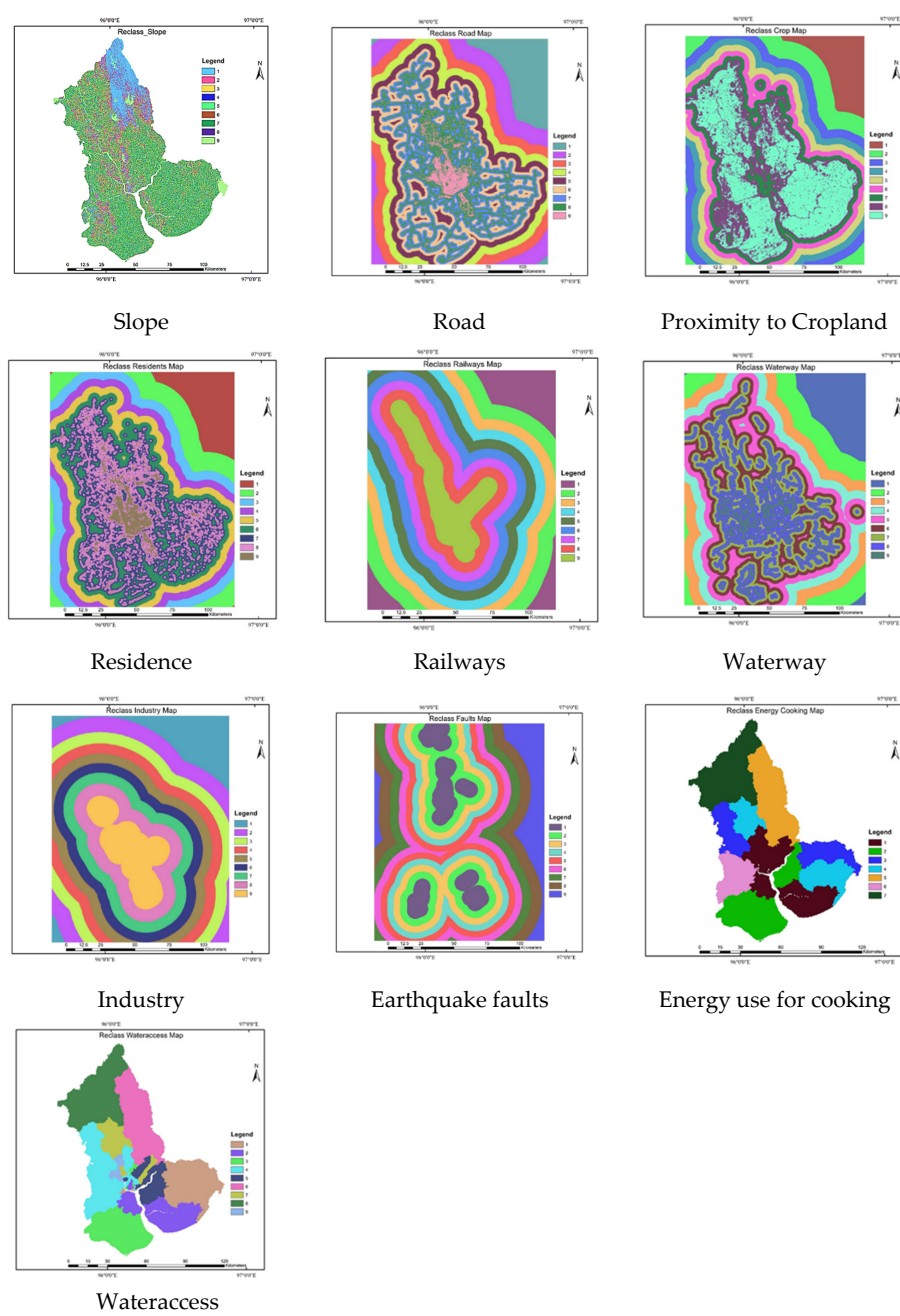

Slope                    Road                    Proximity to Cropland

Residence                    Railways                    Waterway

Industry                    Earthquake faults                    Energy use for cooking

Wateraccess

**Figure A2.** Reclassification of Factor Maps.

## Appendix C

**Table A6.** Crop Residue feedstock (ton/year) in Yangon Region Township.

| Name | Rice Straw | Rice Husk | Ground Nut Straw | Ground Nut Husk | Sesame Trash | Sunflower Residue | Black Gram Crop | Black Gram Hull | Green Gram Crop | Green Gram Hull | Pigeion Crop | Pigeon Hull | Sugarcae Tops | Sugarcane Bagasse | Maize Stalk | Maize Cob |
|---|---|---|---|---|---|---|---|---|---|---|---|---|---|---|---|---|
| Insein | 0.00 | 0.00 | 0.00 | 0.00 | 0.00 | 0.00 | 0.00 | 0.00 | 0.00 | 0.00 | 0.00 | 0.00 | 0.00 | 0.00 | 0.00 | 0.00 |
| Mingaladon | 1207.23 | 183.45 | 0.00 | 0.00 | 0.00 | 0.00 | 0.00 | 0.00 | 0.00 | 0.00 | 0.00 | 0.00 | 0.00 | 0.00 | 0.00 | 0.00 |
| Hmawbi | 11,308.38 | 1718.46 | 452.54 | 93.85 | 0.00 | 0.00 | 323.14 | 5.70 | 13.44 | 0.24 | 0.00 | 0.00 | 0.00 | 0.00 | 0.00 | 0.00 |
| Hlegu | 292,719.14 | 44,482.65 | 4398.90 | 912.29 | 43.08 | 0.00 | 2493.22 | 44.00 | 7115.42 | 125.57 | 0.00 | 0.00 | 0.00 | 0.00 | 0.00 | 0.00 |
| Taikkyi | 170,932.38 | 25,975.50 | 2608.66 | 541.01 | 2086.27 | 25.31 | 29,471.58 | 520.09 | 623.90 | 11.01 | 0.00 | 0.00 | 38,424.00 | 38,295.92 | 5140.65 | 701.70 |
| Htantabin | 355,211.87 | 53,979.26 | 690.63 | 143.23 | 2.15 | 0.00 | 5122.29 | 90.39 | 229.25 | 4.05 | 0.00 | 0.00 | 1594.20 | 1588.89 | 799.30 | 109.10 |
| Shwepyithar | 274.16 | 41.66 | 0.00 | 0.00 | 0.00 | 0.00 | 9.16 | 0.16 | 0.00 | 0.00 | 0.00 | 0.00 | 0.00 | 0.00 | 0.00 | 0.00 |
| Hlaingtharya | 0.00 | 0.00 | 0.00 | 0.00 | 0.00 | 0.00 | 0.00 | 0.00 | 0.00 | 0.00 | 0.00 | 0.00 | 0.00 | 0.00 | 0.00 | 0.00 |
| Thingangyun | 0.00 | 0.00 | 0.00 | 0.00 | 0.00 | 0.00 | 0.00 | 0.00 | 0.00 | 0.00 | 0.00 | 0.00 | 0.00 | 0.00 | 0.00 | 0.00 |
| Yankin | 0.00 | 0.00 | 0.00 | 0.00 | 0.00 | 0.00 | 0.00 | 0.00 | 0.00 | 0.00 | 0.00 | 0.00 | 0.00 | 0.00 | 0.00 | 0.00 |
| South Okkalapa | 0.00 | 0.00 | 0.00 | 0.00 | 0.00 | 0.00 | 0.00 | 0.00 | 0.00 | 0.00 | 0.00 | 0.00 | 0.00 | 0.00 | 0.00 | 0.00 |
| North Okkalapa | 0.00 | 0.00 | 0.00 | 0.00 | 0.00 | 0.00 | 0.00 | 0.00 | 0.00 | 0.00 | 0.00 | 0.00 | 0.00 | 0.00 | 0.00 | 0.00 |
| Thaketa | 0.00 | 0.00 | 0.00 | 0.00 | 0.00 | 0.00 | 0.00 | 0.00 | 0.00 | 0.00 | 0.00 | 0.00 | 0.00 | 0.00 | 0.00 | 0.00 |
| Dawbon | 0.00 | 0.00 | 0.00 | 0.00 | 0.00 | 0.00 | 0.00 | 0.00 | 0.00 | 0.00 | 0.00 | 0.00 | 0.00 | 0.00 | 0.00 | 0.00 |
| Tamwe | 0.00 | 0.00 | 0.00 | 0.00 | 0.00 | 0.00 | 0.00 | 0.00 | 0.00 | 0.00 | 0.00 | 0.00 | 0.00 | 0.00 | 0.00 | 0.00 |
| Pazundaung | 0.00 | 0.00 | 0.00 | 0.00 | 0.00 | 0.00 | 0.00 | 0.00 | 0.00 | 0.00 | 0.00 | 0.00 | 0.00 | 0.00 | 0.00 | 0.00 |
| Botahtaung | 0.00 | 0.00 | 0.00 | 0.00 | 0.00 | 0.00 | 0.00 | 0.00 | 0.00 | 0.00 | 0.00 | 0.00 | 0.00 | 0.00 | 0.00 | 0.00 |
| Dagon Myothit (South) | 0.00 | 0.00 | 0.00 | 0.00 | 0.00 | 0.00 | 0.00 | 0.00 | 0.00 | 0.00 | 0.00 | 0.00 | 0.00 | 0.00 | 0.00 | 0.00 |
| Dagon Myothit (North) | 0.00 | 0.00 | 0.00 | 0.00 | 0.00 | 0.00 | 0.00 | 0.00 | 3.44 | 0.06 | 0.00 | 0.00 | 0.00 | 0.00 | 0.00 | 0.00 |
| Dagon Myothit (East) | 22,458.52 | 3412.88 | 0.00 | 0.00 | 0.00 | 0.00 | 0.00 | 0.00 | 3.44 | 0.06 | 0.00 | 0.00 | 0.00 | 0.00 | 0.00 | 0.00 |
| Dagon Myothit (Seikkan) | 15,501.94 | 2355.73 | 0.00 | 0.00 | 0.00 | 0.00 | 62.35 | 1.10 | 0.00 | 0.00 | 0.00 | 0.00 | 0.00 | 0.00 | 0.00 | 0.00 |
| Mingalar taungnyunt | 0.00 | 0.00 | 0.00 | 0.00 | 0.00 | 0.00 | 0.00 | 0.00 | 0.00 | 0.00 | 0.00 | 0.00 | 0.00 | 0.00 | 0.00 | 0.00 |
| Thanlyin | 157,252.94 | 23,896.72 | 118.43 | 24.56 | 0.00 | 0.00 | 0.00 | 0.00 | 30,077.54 | 530.78 | 0.00 | 0.00 | 0.00 | 0.00 | 0.00 | 0.00 |
| Kyauktan | 262,814.46 | 39,938.23 | 0.00 | 0.00 | 0.00 | 2.26 | 0.00 | 0.00 | 36,466.91 | 643.53 | 0.00 | 0.00 | 0.00 | 0.00 | 0.00 | 0.00 |
| Thongwa | 381,007.22 | 57,899.22 | 139.50 | 28.93 | 0.00 | 2.00 | 0.00 | 0.00 | 109,480.28 | 1932.00 | 0.00 | 0.00 | 0.00 | 0.00 | 0.00 | 0.00 |

**Table A6.** *Cont.*

| Name | Rice Straw | Rice Husk | Ground Nut Straw | Ground Nut Husk | Sesame Trash | Sunflower Residue | Black Gram Crop | Black Gram Hull | Green Gram Crop | Green Gram Hull | Pigeion Crop | Pigeon Hull | Sugarcae Tops | Sugarcane Bagasse | Maize Stalk | Maize Cob |
|---|---|---|---|---|---|---|---|---|---|---|---|---|---|---|---|---|
| Kayan | 277,279.32 | 42,136.36 | 89.96 | 18.66 | 0.00 | 0.54 | 11.38 | 0.20 | 0.00 | 0.00 | 0.00 | 0.00 | 0.00 | 0.00 | 0.00 | 0.00 |
| Twantay | 291,133.19 | 44,241.64 | 0.00 | 0.00 | 0.00 | 0.00 | 170.06 | 3.00 | 5.11 | 0.09 | 1.39 | 0.02 | 0.00 | 0.00 | 16,765.60 | 2288.50 |
| Kawhmu | 261,696.52 | 39,768.34 | 0.00 | 0.00 | 0.00 | 0.54 | 3.28 | 0.06 | 11.60 | 0.20 | 0.00 | 0.00 | 1263.30 | 1259.09 | 0.00 | 0.00 |
| Kungyangon | 285,000.43 | 43,309.68 | 0.00 | 0.00 | 0.00 | 0.00 | 0.00 | 0.00 | 7.53 | 0.13 | 0.00 | 0.00 | 0.00 | 0.00 | 0.00 | 0.00 |
| Dala | 110,021.13 | 16,719.20 | 0.00 | 0.00 | 0.00 | 0.00 | 0.00 | 0.00 | 0.00 | 0.00 | 0.00 | 0.00 | 0.00 | 0.00 | 0.00 | 0.00 |
| Seikgyikanaungto | 701.33 | 106.58 | 0.00 | 0.00 | 0.00 | 0.00 | 0.00 | 0.00 | 0.00 | 0.00 | 0.00 | 0.00 | 0.00 | 0.00 | 0.00 | 0.00 |
| Cocokyun | 0.00 | 0.00 | 0.00 | 0.00 | 0.00 | 0.00 | 0.00 | 0.00 | 0.00 | 0.00 | 0.00 | 0.00 | 0.00 | 0.00 | 0.00 | 0.00 |
| Kyaukta da | 0.00 | 0.00 | 0.00 | 0.00 | 0.00 | 0.00 | 0.00 | 0.00 | 0.00 | 0.00 | 0.00 | 0.00 | 0.00 | 0.00 | 0.00 | 0.00 |
| Pabedan | 0.00 | 0.00 | 0.00 | 0.00 | 0.00 | 0.00 | 0.00 | 0.00 | 0.00 | 0.00 | 0.00 | 0.00 | 0.00 | 0.00 | 0.00 | 0.00 |
| Lanma daw | 0.00 | 0.00 | 0.00 | 0.00 | 0.00 | 0.00 | 0.00 | 0.00 | 0.00 | 0.00 | 0.00 | 0.00 | 0.00 | 0.00 | 0.00 | 0.00 |
| Latha | 0.00 | 0.00 | 0.00 | 0.00 | 0.00 | 0.00 | 0.00 | 0.00 | 0.00 | 0.00 | 0.00 | 0.00 | 0.00 | 0.00 | 0.00 | 0.00 |
| Ahlone | 0.00 | 0.00 | 0.00 | 0.00 | 0.00 | 0.00 | 0.00 | 0.00 | 0.00 | 0.00 | 0.00 | 0.00 | 0.00 | 0.00 | 0.00 | 0.00 |
| Kyeemyn daing | 3637.47 | 552.76 | 0.00 | 0.00 | 0.00 | 0.00 | 0.00 | 0.00 | 0.00 | 0.00 | 0.00 | 0.00 | 0.00 | 0.00 | 0.00 | 0.00 |
| San chaung | 0.00 | 0.00 | 0.00 | 0.00 | 0.00 | 0.00 | 0.00 | 0.00 | 0.00 | 0.00 | 0.00 | 0.00 | 0.00 | 0.00 | 0.00 | 0.00 |
| Hlaing | 0.00 | 0.00 | 0.00 | 0.00 | 0.00 | 0.00 | 0.00 | 0.00 | 0.00 | 0.00 | 0.00 | 0.00 | 0.00 | 0.00 | 0.00 | 0.00 |
| Kamar yut | 0.00 | 0.00 | 0.00 | 0.00 | 0.00 | 0.00 | 0.00 | 0.00 | 0.00 | 0.00 | 0.00 | 0.00 | 0.00 | 0.00 | 0.00 | 0.00 |
| Mayan gone | 0.00 | 0.00 | 0.00 | 0.00 | 0.00 | 0.00 | 0.00 | 0.00 | 0.00 | 0.00 | 0.00 | 0.00 | 0.00 | 0.00 | 0.00 | 0.00 |
| Dagon | 0.00 | 0.00 | 0.00 | 0.00 | 0.00 | 0.00 | 0.00 | 0.00 | 0.00 | 0.00 | 0.00 | 0.00 | 0.00 | 0.00 | 0.00 | 0.00 |
| Bahan | 0.00 | 0.00 | 0.00 | 0.00 | 0.00 | 0.00 | 0.00 | 0.00 | 0.00 | 0.00 | 0.00 | 0.00 | 0.00 | 0.00 | 0.00 | 0.00 |
| Total | 2,900,157.63 | 440,718.32 | 8498.62 | 1762.54 | 2131.50 | 30.67 | 37,666.46 | 664.70 | 184,037.86 | 3247.73 | 1.39 | 0.02 | 41,281.50 | 41,143.90 | 22,705.55 | 3099.31 |

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
