# Peer review of "GIS-Based Cluster and Suitability Analysis of Crop Residues: A Case Study in Yangon Region, Myanmar"

_applsci, doi:10.3390/app122211822_

Round 1

Reviewer 1 Report

This is a valuable study. Overall, the paper is written redundantly. I suggest a major revision.

Comments

Please readjust the resolution of all figures, the words and numbers in the figures are not clear.

Some tables are redundant. Please merge table1 and table2. The contents of Table 9 and Table 11 can be expressed in the text. AHP is a widely used method, table 7 is not necessary. I recommend that the maximum number of tables in the main text be no more than 5 and that the rest be moved to the supplementary material.

Please add the full name for the abbreviation in Fig.2.

Please check the formula in the full text. For example, please add variable names for eq(1)-(2). The characters F, W, and C should be lowercase in eq (4). Please add the definition of symbol i. Also, the numbers of the formulas are discontinuous.

Line 219. Please add full names for PROMETHEE ELECTRE, and TOPSIS Please check the full abbreviation.

Line 292. Please streamline the subheadings. For example, remove (Combination of buffalo, cattle, pig, and chicken manure)

“ArcGIS”, not “Arc GIS”. Please check the full text.

Some sentences are not written properly, please check the language of the whole text.

Author Response

Thank you very much for taking the time to revise our manuscript. Please find the response in the attached file.

Reviewer 2 Report

The paper is about GIS based analysis of crop residues in the selected areas of Yangon. The paper needs a lot of improvements prior to consider it for publication.

Consider first 2-3 lines of the abstract; these should present a clear picture of either research void filled by this study or present the justification of this work.

In introduction, write about the potential of biomass referred in the text for energy generation or other purposes. Also, state the current scenario without GIS based analysis.

The size of the problem, i.e., the amount of residues should be clearly given along with the potential of these residues if applied for value added products.

Line 25: Look for the typo

In text references need corrections

L38-40 Rephrase to give stress over the size of the problem

L41: Correct the formula

L52-54: Either state more than one technique or correct the sentence

L80 Use uniform abbreviations for the terms.

L-96-98 Rephrase to avoid verb confusion

L171-172: Explain the assessment of the data by using two statistical parameters

L192: Replace ‘are’ with ‘were’

L224-225: Correction is needed

Refer tables in parentheses

L304: Remove the error

Author Response

Thank you very much for kindly taking the time to revise our manuscript. Please find the response to your comments/suggestions in the attached file.

Round 2

Reviewer 1 Report

The manuscript has been improved and I recommend accepting it.